

# Identification of iron metabolism-related genes as prognostic indicators for papillary thyroid carcinoma: a retrospective study

Tiefeng Jin[1,2,*], Luqi Ge[3,*], Jianqiang Chen[2], Wei Wang[4], Lizhuo Zhang[2] and Minghua Ge[2,5,6]

[1] Second Clinical Medical College, Zhejiang Chinese Medical University, Hangzhou, Zhejiang, China
[2] Otolaryngology & Head and Neck Center, Cancer Center, Department of Head and Neck Surgery, Zhejiang Provincial People's Hospital, Hangzhou, Zhejiang, China
[3] Department of Pharmacology, College of Pharmaceutical Sciences, Zhejiang University of Technology, Hangzhou, Zhejiang, China
[4] Department of Pathology, Laboratory Medicine Center, Zhejiang Provincial People's Hospital, Hangzhou, Zhejiang, China
[5] Clinical Research Center for Cancer of Zhejiang Province, Hangzhou, Zhejiang, China
[6] Key Laboratory of Endocrine Gland Diseases of Zhejiang Province, Hangzhou, Zhejiang, China
* These authors contributed equally to this work.

Corresponding authors
Lizhuo Zhang, zlz2020@foxmail.com
Minghua Ge,
geminghua@hmc.edu.cn

## ABSTRACT

**Background:** The thyroid cancer subtype that occurs more frequently is papillary thyroid carcinoma (PTC). Despite a good surgical outcome, treatment with traditional antitumor therapy does not offer ideal results for patients with radioiodine resistance, recurrence, and metastasis. The evidence for the connection between iron metabolism imbalance and cancer development and oncogenesis is growing. Nevertheless, the iron metabolism impact on PTC prognosis is still indefinite.

**Methods:** Herein, we acquired the medical data and gene expression of individuals with PTC from The Cancer Genome Atlas (TCGA) and the Gene Expression Omnibus (GEO) database. Typically, three predictive iron metabolism-related genes (IMRGs) were examined and employed to build a risk score (RS) model *via* the least absolute shrinkage and selection operator (LASSO) regression, univariate Cox, and differential gene expression analyses. Then we analyzed somatic mutation and immune cell infiltration among RS groups. We also validated the prognostic value of two IMRGs (SFXN3 and TFR2) by verifying their biological function through *in vitro* experiments.

**Results:** Based on RS, all patients with PTC were stratified into low- and high-risk groups, where Kaplan-Meier analysis indicated that disease-free survival (DFS) in the high-risk group was much lower than in the low-risk group ($P < 0.0001$). According to ROC analysis, the RS model successfully predicted the 1-, 3-, and 5-year DFS of individuals with PTC. Additionally, in the TCGA cohort, a nomogram model with RS was developed and exhibited a strong capability to anticipate PTC patients' DFS. In the high-risk group, the enriched pathological processes and signaling mechanisms were detected utilizing the gene set enrichment analysis (GSEA). Moreover, the high-risk group had a significantly higher level of BRAF mutation frequency, tumor mutation burden, and immune cell infiltration than the low-risk

group. *In vitro* experiments found that silencing SFXN3 or TFR2 significantly reduced cell viability.

**Conclusion:** Collectively, our predictive model depended on IMRGs in PTC, which could be potentially utilized to predict the PTC patients' prognosis, schedule follow-up plans, and provide potential targets against PTC.

# INTRODUCTION

The most common endocrine malignancy identified is thyroid carcinoma (TC), and recently its frequency has been steadily rising (*Ferrari et al., 2020*). Papillary thyroid carcinoma (PTC), the most prevalent pathological TC type, accounts for 85% of cases that are detected (*Lim et al., 2017*). Although PTC has a good overall prognosis, it has been reported that between 5% and 21% of cases have a recurrent tumor, lymph node metastasis, or distant postoperative metastasis, resulting in a significant reduction in survival rate (*Nixon et al., 2014*). Therefore, it is crucial to clarify the pathogenesis of PTC and detect effective prognostic markers.

For cells to keep functioning and maintain homeostasis, iron is a crucial component. Tumor incidence, progression, and metastasis are closely correlated with iron metabolism imbalance (*Fischer-Fodor et al., 2015*; *Leftin et al., 2017*; *Steegmann-Olmedillas, 2011*). Markedly, it is informed that iron metabolism has two-fold impacts on cancer cells. On one side, iron addiction refers to the phenomenon in which tumor cells, compared to healthy cells, become more dependent on iron intake for development and are more prone to deplete iron (*Dev & Babitt, 2017*). Alternatively, iron overload in tumor cells causes a unique type of programmed cell death named ferroptosis, where lipid peroxidation and excessive reactive oxygen species (ROS) cause cell death (*Chen et al., 2021*; *Dixon et al., 2012*). Several studies have identified that iron metabolism-related genes (IMRGs) are important predictors of malignancy prognosis. Nevertheless, the connection between IMRGs and PTC patients' prognosis has not been well examined (*Li et al., 2020*; *Xu et al., 2021*; *Yuan, Liu & Zhang, 2021*).

In this investigation, IMRGs were investigated in PTC. Depending on transcriptome and medical data available from the public database, we conducted extensive bioinformatics studies. We first defined the iron metabolism-related differentially expressed genes (IMR-DEGs) and conducted Pearson's correlation and network analyses. Subsequently, we developed three predictive IMRG signatures, assessed them, and verified them using the least absolute shrinkage and selection operator regression (LASSO, Tibshirani) regression and Cox regression analyses. Using this information as a foundation, we established a risk score (RS) system for PTC, performed multivariate and univariate Cox regression analyses, and then built and assessed a nomogram to anticipate the prognosis. We also assessed somatic mutation and immune cell infiltration among RS

groups. Finally, we performed *in vitro* experiments to validate the function of two candidate genes.

## MATERIALS AND METHODS

### Data download and preprocessing

The Cancer Genome Atlas (TCGA) database was employed to acquire the transcriptome data (RNA-seq) of 494 PTC and 59 normal tissue (NT) samples in conjunction with medical features and follow-up outcome data. Consequently, the tumor group was randomly distributed into model training and validation sets at a 7:3 ratio. Finally, 350 cases of tumor tissue were in the training set, and 144 cases of tumor tissue were in the validation set. Subsequent analyses and prognostic model construction used a training set and validated it in a validation set. At the same time, we downloaded the original microarray outcomes (CEL format) of transcriptome dataset GSE33630 of PTC and NT samples with reliable sample sources from the Gene Expression Omnibus (GEO) database (*Dom et al., 2012*), as well as the annotation file of GPL570[HG-U133_Plus_2] Affymetrix Human Genome U133 Plus 2.0 Array platform. Transcriptome data for 45 NT and 49 PTC were included in the GSE33630 collection.

### Expression and correlation analysis of IMRGs

IMRGs were from two databases: Iron uptake and transport (R-HAS-917937) pathway genes from the Reactome database (https://reactome.org/) and cellular iron ion homeostasis genes (GO:0006879) from the AmiGo2 database (http://amigo.geneontology.org/amigo). Finally, 70 IMRGs were screened (Table 1). The IMR-DEGs were identified utilizing the limma package of R language (*Ritchie et al., 2015*) using a linear model. Adj. $P < 0.05$ and $|log2FC| > 1$ were the criteria for DEG screening. We used the ggplot2 package (*Villanueva & Chen, 2019*) and pheatmap package (*Kolde, 2019*) to the heatmaps and volcano plots of differentially expressed genes (DEGs) to illustrate the differential expression analysis outcomes. Pearson's correlation was then analyzed between different IMR-DEGs.

### Protein-protein interaction (PPI) network analysis and transcription factor, miRNA, drug-protein interaction network analysis

For IMR-DEGs, we built a PPI network utilizing the STRING (http://www.string-db.org/, Version: 11.0) online method (*Szklarczyk et al., 2019*) and Cytoscape program (*Otasek et al., 2019*). We further analyzed transcription factors, miRNA, small molecular compounds, and their drug interaction networks. The TF-gene, Gene-miRNA, Protein-chemical, and Protein-drug interactions modules of NetworkAnalyst were used to examine the uploaded DEGs (https://www.networkanalyst.ca/) (*Xia, Gill & Hancock, 2015*), respectively. TF databases, ENCODE (http://cistrome.org/BETA/), miRTarBase v8.0 (https://mirtarbase.cuhk.edu.cn), comparative toxicogenomics database (CTD, http://ctdbase.org/) and DrugBank database v5.0 (https://go.drugbank.com/) were the reference database. The analyzed network was visualized by Cytoscape software.

**Table 1 Information of IMRGs.**

| Gene symbol | Ensemble | Description |
| --- | --- | --- |
| NUBP1 | ENSG00000103274 | Nucleotide-binding protein 1 |
| FBXL5 | ENSG00000118564 | F-box and leucine-rich repeat protein 5 |
| TMEM199 | ENSG00000244045 | Transmembrane protein 199 |
| HJV | ENSG00000168509 | Hemojuvelin BMP co-receptor |
| ALAS1 | ENSG00000023330 | 5′-aminolevulinate synthase 1 |
| ALAS2 | ENSG00000158578 | 5′-aminolevulinate synthase 2 |
| ISCU | ENSG00000136003 | Iron-sulfur cluster assembly enzyme |
| SLC22A17 | ENSG00000092096 | Solute carrier family 22 member 17 |
| FXN | ENSG00000165060 | Frataxin |
| GLRX3 | ENSG00000108010 | Glutaredoxin 3 |
| SLC40A1 | ENSG00000138449 | Solute carrier family 40 member 1 |
| ABCB8 | ENSG00000197150 | ATP binding cassette subfamily B member 8 |
| BDH2 | ENSG00000164039 | 3-hydroxybutyrate dehydrogenase 2 |
| BMP6 | ENSG00000153162 | Bone morphogenetic protein 6 |
| CP | ENSG00000047457 | Ceruloplasmin |
| HEPH | ENSG00000089472 | Hephaestin |
| LTF | ENSG00000012223 | Lactotransferrin |
| NDFIP1 | ENSG00000131507 | Nedd4 family interacting protein 1 |
| LCN2 | ENSG00000148346 | Lipocalin 2 |
| HAVCR1 | ENSG00000113249 | Hepatitis A virus cellular receptor 1 |
| SCARA5 | ENSG00000168079 | Scavenger receptor class A member 5 |
| FLVCR1 | ENSG00000162769 | Feline leukemia virus subgroup C cellular receptor 1 |
| ABCG2 | ENSG00000118777 | ATP binding cassette subfamily G member 2 |
| CYBRD1 | ENSG00000071967 | Cytochrome b reductase 1 |
| SLC48A1 | ENSG00000211584 | Solute carrier family 48 member 1 |
| SLC46A1 | ENSG00000076351 | Solute carrier family 46 member 1 |
| LRP1 | ENSG00000123384 | LDL receptor related protein 1 |
| CD163 | ENSG00000177575 | CD163 molecule |
| FLVCR2 | ENSG00000119686 | Feline leukemia virus subgroup C cellular receptor family member 2 |
| SLC39A14 | ENSG00000104635 | Solute carrier family 39 member 14 |
| SLC39A8 | ENSG00000138821 | Solute carrier family 39 member 8 |
| SLC11A2 | ENSG00000110911 | Solute carrier family 11 member 2 |
| SLC25A37 | ENSG00000147454 | Solute carrier family 25 member 37 |
| HPX | ENSG00000110169 | Hemopexin |
| ERFE | ENSG00000178752 | Erythroferrone |
| TMPRSS6 | ENSG00000187045 | Transmembrane serine protease 6 |
| HAMP | ENSG00000105697 | Hepcidin antimicrobial peptide |
| SLC11A1 | ENSG00000018280 | Solute carrier family 11 member 1 |
| EIF2AK1 | ENSG00000086232 | Eukaryotic translation initiation factor 2 alpha kinase 1 |
| STEAP2 | ENSG00000157214 | STEAP2 metalloreductase |
| HFE | ENSG00000010704 | Homeostatic iron regulator |

| Gene symbol | Ensemble | Description |
| --- | --- | --- |
| CUBN | ENSG00000107611 | Cubilin |
| TFRC | ENSG00000072274 | Transferrin receptor |
| TFR2 | ENSG00000106327 | Transferrin receptor 2 |
| HMOX1 | ENSG00000100292 | Heme oxygenase 1 |
| HMOX2 | ENSG00000103415 | Heme oxygenase 2 |
| STEAP1 | ENSG00000164647 | STEAP family member 1 |
| STEAP3 | ENSG00000115107 | STEAP3 metalloreductase |
| STEAP4 | ENSG00000127954 | STEAP4 metalloreductase |
| ABCB7 | ENSG00000131269 | ATP binding cassette subfamily B member 7 |
| ABCB10 | ENSG00000135776 | ATP binding cassette subfamily B member 10 |
| ABCB6 | ENSG00000115657 | ATP binding cassette subfamily B member 6 |
| SFXN1 | ENSG00000164466 | Sideroflexin 1 |
| SFXN2 | ENSG00000156398 | Sideroflexin 2 |
| SFXN3 | ENSG00000107819 | Sideroflexin 3 |
| SFXN4 | ENSG00000183605 | Sideroflexin 4 |
| SFXN5 | ENSG00000144040 | Sideroflexin 5 |
| ATP6AP1 | ENSG00000071553 | ATPase H+ transporting accessory protein 1 |
| SLC25A28 | ENSG00000155287 | Solute carrier family 25 member 28 |
| ACO1 | ENSG00000122729 | Aconitase 1 |
| IREB2 | ENSG00000136381 | Iron-responsive element binding protein 2 |
| FTH1 | ENSG00000167996 | Ferritin heavy chain 1 |
| FTL | ENSG00000087086 | Ferritin light chain |
| NCOA4 | ENSG00000266412 | Nuclear receptor coactivator 4 |
| PCBP1 | ENSG00000169564 | Poly(rC) binding protein 1 |
| PCBP2 | ENSG00000197111 | Poly(rC) binding protein 2 |
| FTHL17 | ENSG00000132446 | Ferritin heavy chain like 17 |
| TF | ENSG00000091513 | Transferrin |
| FTMT | ENSG00000181867 | Ferritin mitochondrial |
| FECH | ENSG00000066926 | Ferrochelatase |

## Screening of prognostic markers

For the screened IMR-DEGs, we used Cox regression analysis to assess the connection between disease-free survival (DFS) of individuals with PTC and gene expression. A compression estimation is the LASSO approach (*Tibshirani, 1996*). By designing a penalty function, which causes it to compress certain coefficients and set others to zero, the model is refined. As a result, it still has the benefit of subset shrinking and is biased when processing data that have complicated collinearity. Considering the findings of survival analysis, we employed LASSO regression to further identify predictive biomarkers. The variables were screened using the glmnet function of the glmnet package (*Engebretsen*

*& Bohlin, 2019*) and cross-validated using the cv. glmnet function, and then the combination of predictive biomarkers with the lowest CV coefficient was selected.

## Construction of RS and evaluation of clinical prognosis predictive ability

The following formula was employed to detect RS for each case:

$$RS = \sum_{i=1}^{n} \text{Coef}_i \times Exp_i,$$

where Coef was the LASSO regression coefficient and Exp was the gene expression level (log2 conversion).

The best RS cutoff value for predicting the survival time of individuals with PTC was estimated utilizing the maxstat package (*Hothorn, 2017*). Subsequently, according to the cutoff value, participants were divided into low- and high-risk groups, and the survival curve was plotted utilizing the Kaplan-Meier technique. Simultaneously, the survival period of 1-, 3- and 5-year for individuals with RS was predicted utilizing the survivalROC package (*Heagerty, 2013*). Then the anticipated ROC was mapped, and the AUC value was estimated.

We used a Cox proportional hazards model to evaluate the other medical features' impact on patients' prognostic survival, including age, sex, tumor stage, and TNM stage, and developed forest maps with the forestmodel package (*Kennedy, 2020*). Subsequently, to evaluate the independent RS predictive value on patient prognosis, the clinical parameters that significantly affected prognosis were included in multivariate Cox regression as covariates. A forest map was then created. The fitting effect of different models was evaluated by AIC values.

Finally, to display the model outcomes and make the prediction model results more readable, the nomogram and calibration curve of the best multifactor model was created utilizing the rms package (*Harrell, 2021*). The nomograms' predictive ability for survival was estimated through the calculation of the concordance index (C-index).

## Differential expression analysis and functional enrichment analysis of risk groups

To investigate the potential RS biological importance, we conducted a differential study of gene expression in various risk groups. Adj. $P < 0.05$ and $|log2FC| > 1$ were the DEG examining criteria. The heatmap for the DEGs and the volcano plot were created utilizing ggplot2 (*Villanueva & Chen, 2019*) and pheatmap packages (*Kolde, 2019*) to show the differential analysis outcomes.

A famous database that includes information on biological mechanisms, genomes, diseases, and medicines is called the Kyoto Encyclopedia of Genes and Genomes (KEGG). Go function annotation analysis, which includes molecular function (MF), cell component (CC), and biological process (BP), is a typical approach for large-scale gene function enrichment study. KEGG enrichment analyses and gene ontology (GO) (*Harris et al., 2004*; *Kanehisa & Goto, 2000*) were conducted on DEGs of risk groups utilizing the clusterProfiler package (*Yu et al., 2012*); $P < 0.05$ was deemed statistically significant.

## Analysis of somatic mutation and immune cell infiltration

Fisher's exact test was performed on all genes in the high-/low-risk group of the TCGA dataset to detect differentially mutated genes using the "mafCompare" function in the R package Maftools (version: 2.14.0) (*Mayakonda et al., 2018*). Subsequently, the Wilcoxon test was used to compare tumor mutation burden (TMB) between high- and low-risk groups, and a violin plot was used to present the results.

We used the gene expression matrix from the TCGA database in the CIBERSORTx (https://cibersortx.stanford.edu/) (*Newman et al., 2015*) online analysis tool to calculate the immune cell infiltration of the samples and filtered out samples with $P < 0.05$.

## The patients and the tissue samples

We collected PTC samples retrospectively from patients, who had undergone one-stage surgery to analyze the expression of SFXN3 or TFR2. A written informed consent form was completed by each patient, and all studies were approved by the ethics committee of Zhejiang provincial people's hospital (Ethical Approval No: 2021QT251).

## Immunohistochemistry (IHC)

Xylene was employed to deparaffinized paraffin-embedded thyroid segments, and ethanol was utilized to rehydrate it. The antigen was then recovered from the segments utilizing 1 mM EDTA, and non-specific binding was reduced by preincubating the segments in TBS with 5% goat serum. The sections were then treated with either the TFR2 (A9845; ABclonal, Wuhan, China) or SFXN3 (15156-1-AP; Proteintech, San Diego, CA, USA) antibodies. Thyroid tissue samples were examined using DAB and counterstained with hematoxylin after treatment with horseradish peroxidase secondary antibodies.

Every specimen's immune-positive rate and staining levels were incorporated into the IHC score. Three pathologists examined all the slices' protein-expressing scores in the current investigation. SFXN3 and TFR2's immunoreactivity score (IRS), which ranged from 0 to 12, was determined as the intensity and positive rate product (*Song et al., 2021*).

## Cell culture

The PTC cell line TPC-1 was given by Dr. Xin Zhu (Zhejiang Cancer Hospital). It was cultured in RPMI-1640 + 10% FBS. Short tandem repeat (STR) profiling has been employed to regularly detect mycoplasma in all cell lines and to verify their authenticity.

## Transfection

SiRNAs (small interfering RNA) targeting SFXN3 and TFR2 were obtained from RiboBio (RiboBio Biotechnology, Guangzhou, China), the sequence of SFXN3 siRNA#1 was CTGGAAGCTTCTCGGAACA, the sequence of SFXN3 siRNA#2 was GCGTTGAAGTGGTCTACTA, the sequence of SFXN3 siRNA#3 was GGTGAATTGCCTTTAGACA, the sequence of TFR2 siRNA#1 was GGCTAGTGGTCAACAATCA, the sequence of TFR2 siRNA#2 was CTCAAGGAGTGCTCATATA, and the sequence of TFR2 siRNA#3 was TGGGCAGACTCTCTATGAA. We transfected SFXN3-siRNAs or TFR2-siRNAs

**Table 2 Primers for the different genes.**

| Gene ID | Forward primer | Reverse primer |
|---------|----------------|----------------|
| SFXN3 | TCGCTGGGACCAAAGTACTT | GGATGGAAGGCGGAGTCATA |
| TFR2 | GTGTTCACCAATCCCAGCTG | GTGTCCTCCTTTGTGTGCAG |
| TFR | GAAAACAGACAGATTTGTCATG | CTCTTTTGGAGATACGTAGGG |
| DMT1(+)IRE | TGGCTTATCTGGGCTTTGTG | CACACTGGCTCTGATGGCTA |
| DMT1(−)IRE | GTGGCATTATATGTGGTGGC | CAGCGTCCATGGTGTTCAGA |
| FPN | AACAAGCACCTCAGCGAGAG | CACATCCGATCTCCCCAAGT |
| FTL | GGCCCTGGAGAAAAAGC | GAAGTGAGTCTCCAGGAAG |
| FTH | CAGATCAACCTGGAGCTCTAC | CTTCAAAGCCACATCATCGC |
| GAPDH | GCACCGTCAAGGCTGAGAAC | TGGTGAAGACGCCAGTGGA |

using jetPRIME (Polyplus, Berkeley, CA, USA) at final doses of 50 nM after cells had been grown in a six-well plate at 40% confluence. After 2 days, cells were obtained for further trials.

## qRT-PCR and western blots (WB)

We performed qRT-PCR and WB as previously described (*Pan et al., 2022*). Table 2 displayed the primers used for each gene.

For WB, the primary antibodies included: ferritin light chain (FTL, 10727-1-AP; Proteintech, San Diego, CA, USA), ferritin heavy chain (FTH, YT1692; ImmunoWay, Jiangsu, China), transferrin receptor (TFR, YT5374; ImmunoWay, Jiangsu, China), GAPDH (10494-1-AP, Proteintech, San Diego, CA, USA).

## Cell viability assay

For cell viability assays, cells ($2 \times 10^3$ cells/well) transfected with siRNA were incubated for two days in 96-well plates. Cultures were examined utilizing the Cell Counting Kit-8 (CCK-8; Beyotime, Shanghai, China) following the manufacturer's directions.

## Statistical analyses

The R program (*R Core Team, 2022*) was employed to perform all statistical analysis and data calculations. The Benjamini-Hochberg (BH) method was utilized for multiple test corrections, and FDR correction was performed in multiple tests to decrease the false positive rate. For the comparison of two groups, the Mann-Whitney U test (*i.e.*, Wilcoxon rank sum test) was employed to compare the differences of continuous variables with non-normal distributions, while the independent student T-test was utilized to determine the statistical significance of continuous variables with normal distributions. To evaluate the prognostic biomarkers' predictive value, a Cox regression model was used. The pROC package of R was employed to plot the receiver operating characteristic (ROC) curve, and the accuracy of RS in determining prognosis was evaluated by computing the area under the curve (AUC). *P*-values for all tests were two-sided, with $P < 0.05$ indicating statistical significance.

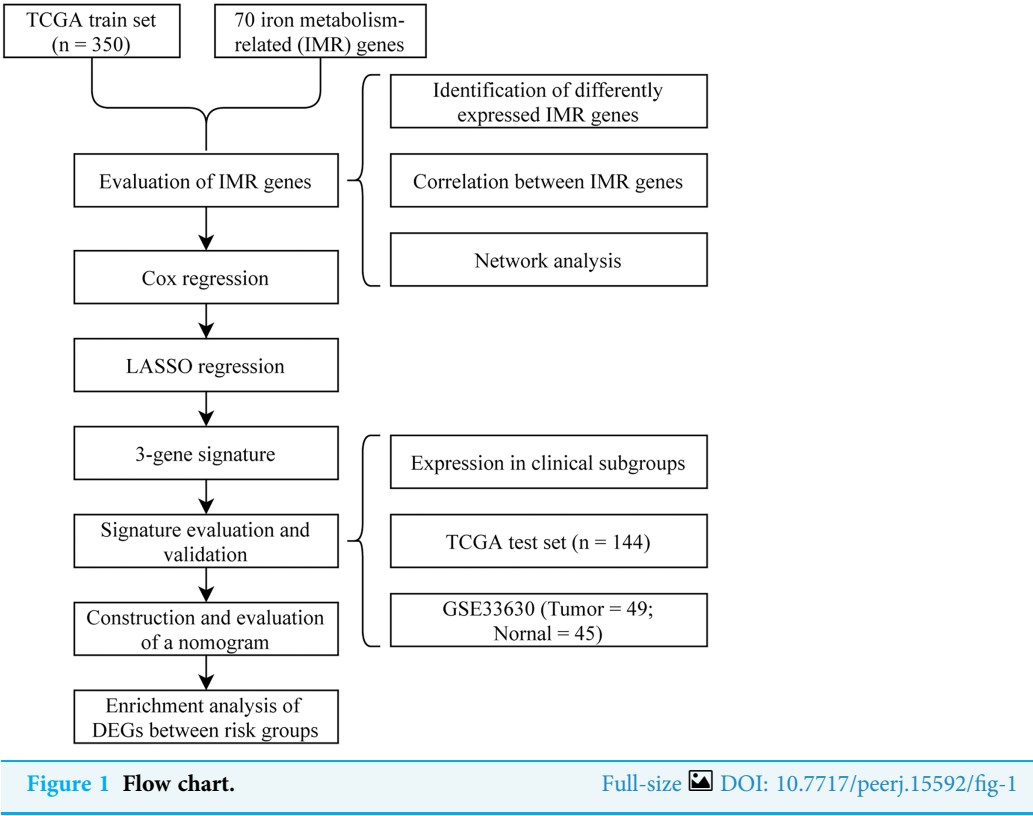

**Figure 1 Flow chart.**

## RESULTS

To visually and systematically describe our work, we presented the research process in Fig. 1.

### Differential expression analysis and correlation analysis of IMRGs

The 70 IMRGs expression levels in the tumor and control groups were compared. Finally, we discovered that the 12 IMRGs expression had significant differences, among which SFXN3, TFR2, TMPRSS6, HMOX1, and LCN2 expressions were elevated in the cancer group, and SCARA5, LTF, STEAP2, SLC39A14, CP, ALAS2, and TF were low expressed in the tumor group (Fig. 2A). The IMRGs expression levels in different samples were displayed by heatmap (Fig. 2B). Figure 2C illustrates the correlation matrix of IMRGs expression level. Among them, HMOX1 exhibited significant positive correlations with TMPRSS6, SFXN3, and LCN2 ($P < 0.001$); SFXN3 exhibited significant positive correlations with TMPRSS6, LCN2, and TFR2 ($P < 0.001$); LCN2 exhibited a significant positive correlation with TMPRSS6 ($P < 0.001$); SCARA5 exhibited a significant positive correlation with CP ($P < 0.001$). Each correlation coefficient was greater than or equal to 0.45. In contrast, SLC39A14 exhibited a significant negative correlation with TMPRSS6 ($P < 0.001$), with a correlation coefficient of less than −0.45 (Fig. 3).

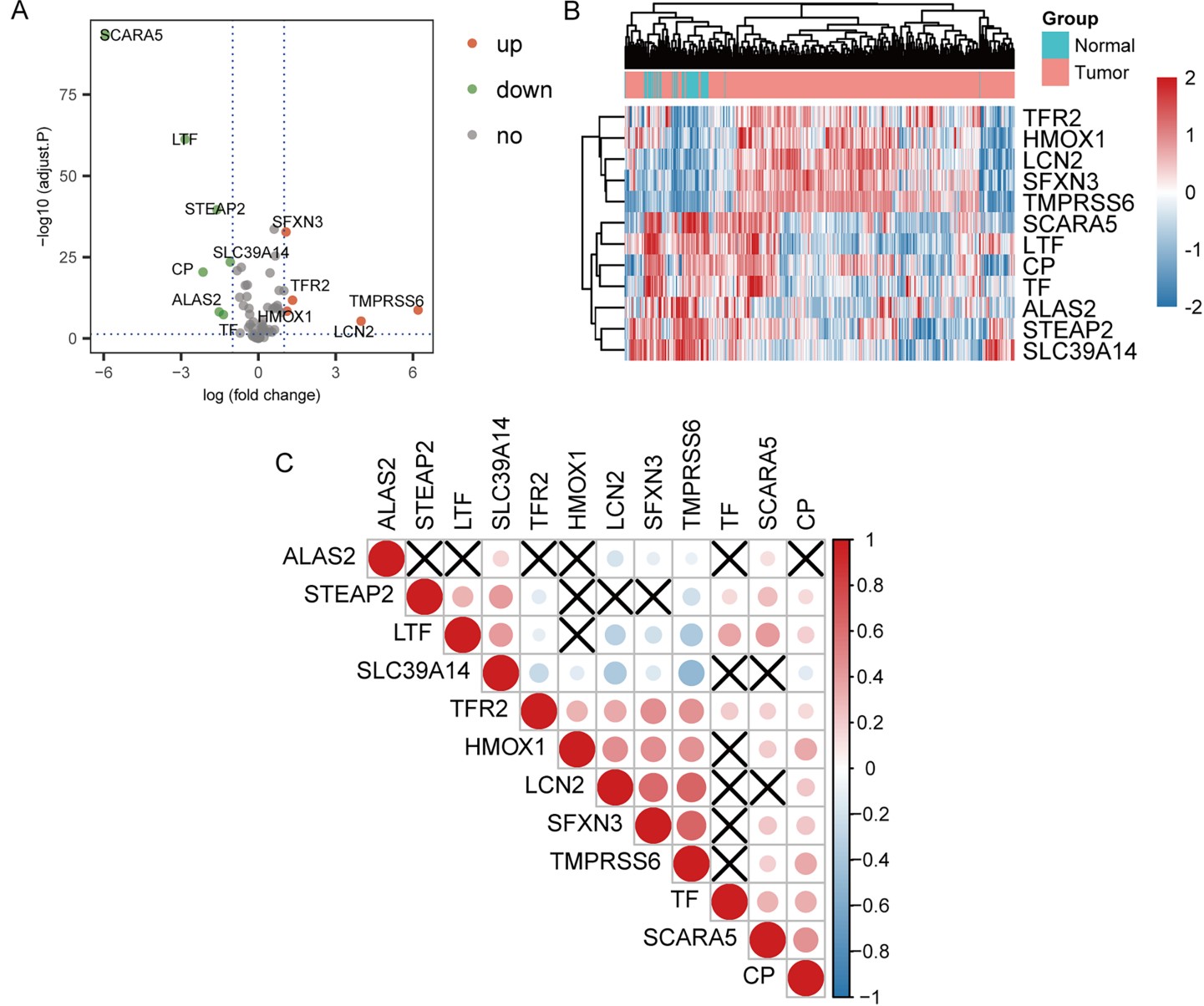

**Figure 2 Expression and correlation analysis of IMRGs.** (A) The volcano plot of comparison of IMRGs expression levels between the tumor and the control groups. The green and orange dots represented the down-regulated and up-regulated genes, respectively. (B) The IMR-DEGs expression level in different samples is shown in a heatmap. (C) The correlation matrix of IMR-DEG expression levels. Blue revealed a negative association, and red revealed a positive association. The darker the color, the greater the degree of correlation. Those without statistical significance were indicated in black.

## Protein, transcription factor, miRNA, small molecule compound, and drug interaction networks of IMR-DEGs

PPI network analysis of IMR-DEGs was conducted in cancer and healthy tissues, and Fig. 4A presents the PPI network. In the PPI network, TFR2, CP, and SLC39A14 had a large weight and a strong connection. The transcription factor analysis showed that SFXN3 was related to 72 transcription factors, LCN2 was related to 45 transcription factors, HMOX1 was related to 42 transcription factors, SLC39A14 was related to 35

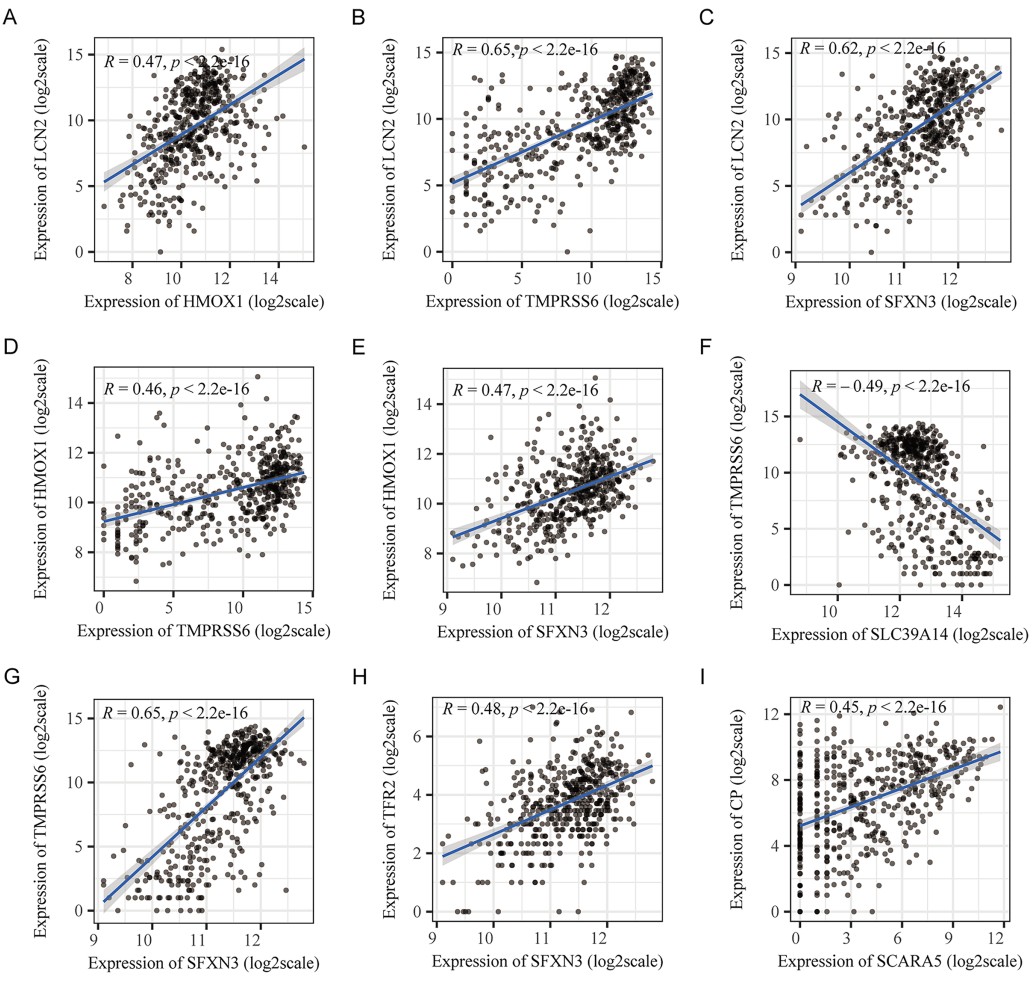

**Figure 3 Pearson's correlation analysis point line diagram of IMRGs.** (A–I) The significant corre-
lation analysis outcomes of IMRGs expression level, respectively, and the point line diagram of results
were shown with correlation coefficients more than 0.45.

transcription factors, CP was related to 24 transcription factors, TF was related to 24
transcription factors, and STEAP2 was related to 11 transcription factors. The miRNA
interaction network analysis showed that HMOX1 was related to 47 miRNAs, STEAP2
was related to 43 miRNAs, SLC39A14 was related to 22 miRNAs, SFXN3 was related to
20 miRNAs, LTF was related to 18 miRNAs, and TMPRSS6 was related to 14 miRNA.
The results of small molecule compound analysis showed that HMOX1 was related to 545
compounds, TF was related to 89 compounds, CP was related to 80 compounds, LCN2
was related to 41 compounds, SLC39A14 was related to 39 compounds, LTF was related
to 33 compounds, STEAP2 was related to 25 compounds, and SFXN3 was related to 22
compounds. The results of the drug interaction analysis showed that LCN2 was related to
six drugs, HMOX1 was related to nine drugs, and ALAS2 was related to two drugs
(Fig. 4).

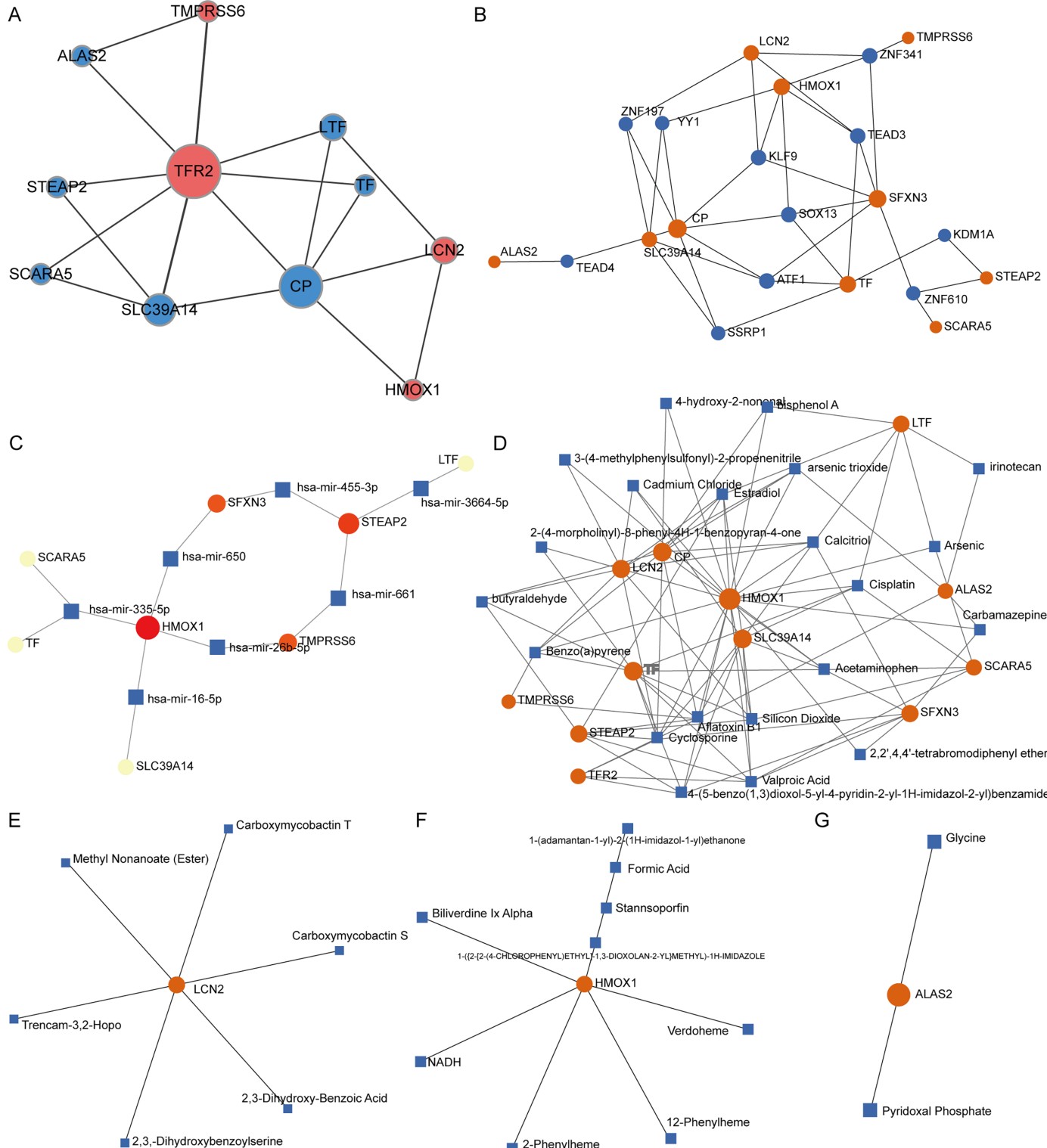

**Figure 4** **Interaction network analysis of IMR-DEGs.** (A) The PPI network of 12 IMR-DEGs, blue for down-regulated genes, red for up-regulated genes, and the bigger the circle, the greater the network weight. (B) The interaction network of IMR-DEGs and transcription factors, the core network was displayed after the source network was simplified by the minimum network algorithm. (C) The interaction network of IMR-DEGs and miRNA, the core network was displayed after the source network was simplified by the minimum network algorithm. (D) The interaction network of IMR-DEGs and small molecular compounds, the core network was displayed after the source network was simplified by the minimum network algorithm. (E–G) The interaction network of IMR-DEGs and drugs.

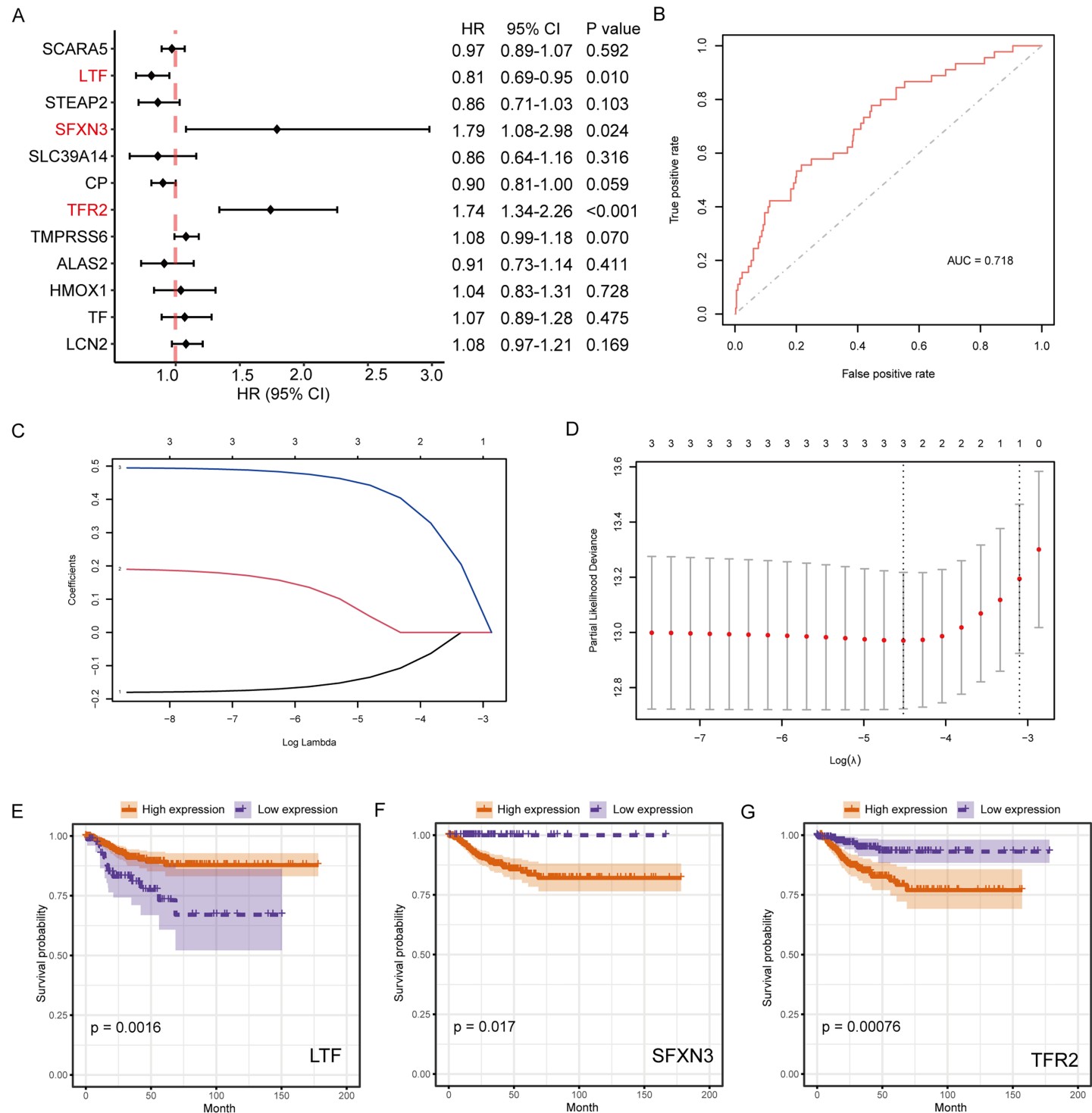

**Figure 5 Survival analysis of IMRGs and LASSO regression was employed to examine predictive markers.** (A) Cox regression analysis of IMRGs expression level impact on the patient's prognosis, displayed in the forest map. (B) The fitting result of LASSO regression used the ROC curve to assess the predictive ability of prognosis, with AUC as the area under the curve. (C and D) The LASSO Cox regression model was employed to screen predictive markers, and the incomplete likelihood deviation with 10 times cross-validation was utilized to calculate the best λ. (E–G) The high-/low-expression groups' survival curves (grouped by median) with significant results of Cox analysis, respectively.

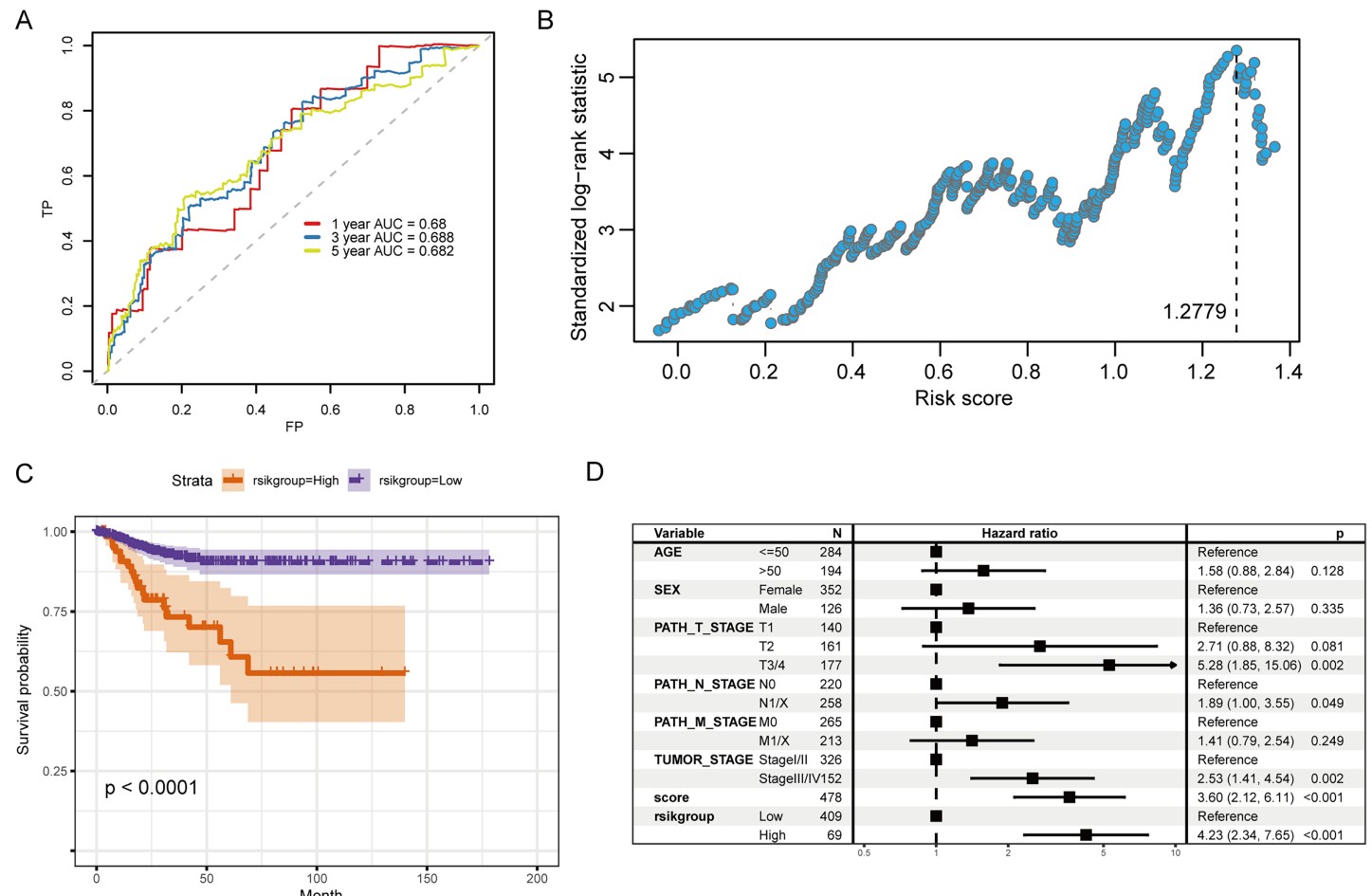

**Figure 6 Evaluation of the prognostic ability of RS for prognosis survival time of PTC patients.** (A) ROC curve and calculated AUC value of RS for predicting 1-, 3- and 5-year survival. (B) The calculation dot plot of the best cutoff value of RS, with the cutoff value marked by a dotted line. (C) Survival curve for high/low expression groups of RS (K–M method). (D) Cox regression analysis of the RS, grouping, and other medical characteristics effects on the prognosis of patients, presented as a forest map.

## Survival analysis of IMRGs and screening of prognostic markers

The connection between IMR-DEGs and the patient's prognosis was examined utilizing univariate Cox regression, and it found that LTF, SFXN3, and TFR2 were significantly connected with prognosis (Fig. 5A). Subsequently, LASSO regression analysis was performed and three IMRGs were still retained, which could be used as joint prognostic markers (Figs. 5B–5D). The prognostic ROC curve for the LASSO regression revealed an AUC value of 0.718, which indicates good prediction ability (Fig. 5B). Figures 5E–5G show the survival curves for the three prognostic markers grouped by high-/low-expression.

## RS and prognostic predictive model construction

The coefficients of candidate predictive biomarkers were detected based on the LASSO regression model outcomes, and the RS was detected with the following formula:

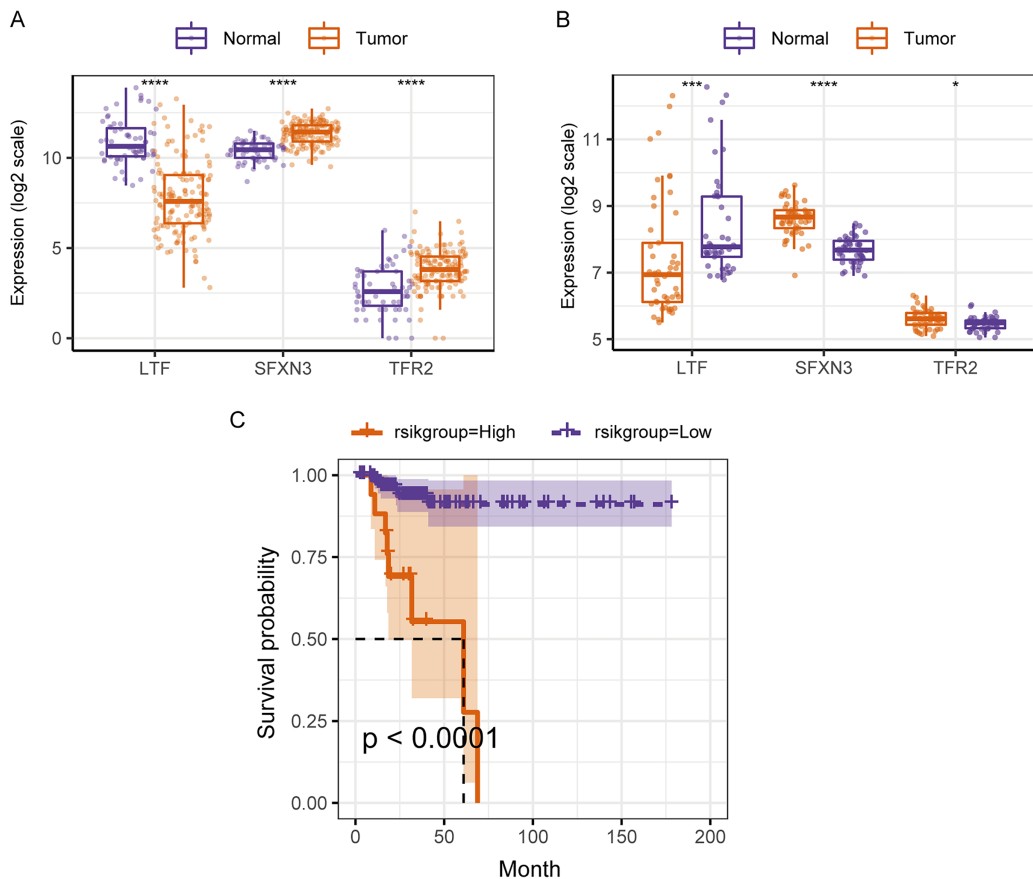

**Figure 7 Assessment and validation of the predictive ability of candidate prognostic markers and RS for clinical prognosis of subjects with PTC.** (A) The three prognostic markers' expression differences were analyzed in the TCGA validation set between PTC and normal tissues. Kruskal-Wallis test was employed to examine the variations between groups, and asterisks (*$P < 0.05$, ***$P < 0.001$, ****$P < 0.0001$) indicated the statistically significant differences. (B) Differential expression analysis of three prognostic markers between PTC and healthy tissues in the GEO dataset GSE33630. (C) Survival curve of high-/low- expression groups of RS in TCGA validation set (K–M method).

$$RS = (-0.1201) * LTF + (0.0053) * SFXN3 + (0.4250) * TFR2.$$

Figure 6A illustrates the ROC curve indicated by the score for the 1-, 3- and 5-year survival. Among them, the predictive ability for the 3-year survival was the best (AUC = 0.688). We then used the maxstat package to determine that the best RS threshold value for anticipating the survival time of individuals with PTC was 1.2779 (Fig. 6B). According to the cutoff value, we distributed the cases into high- and low-risk groups and patients with increased RS had a significantly lower prognostic survival time than those with low RS (Fig. 6C). The univariate COX regression showed that besides the RS/ grouping, cancer, N, and T stages had an influence on the patient's survival (Fig. 6D).

Therefore, in the validation set, we validated the predictive biomarkers examined by LASSO regression. The TCGA validation set indicated that differential expression was observed between tumor and control groups for the three prognostic markers, and the

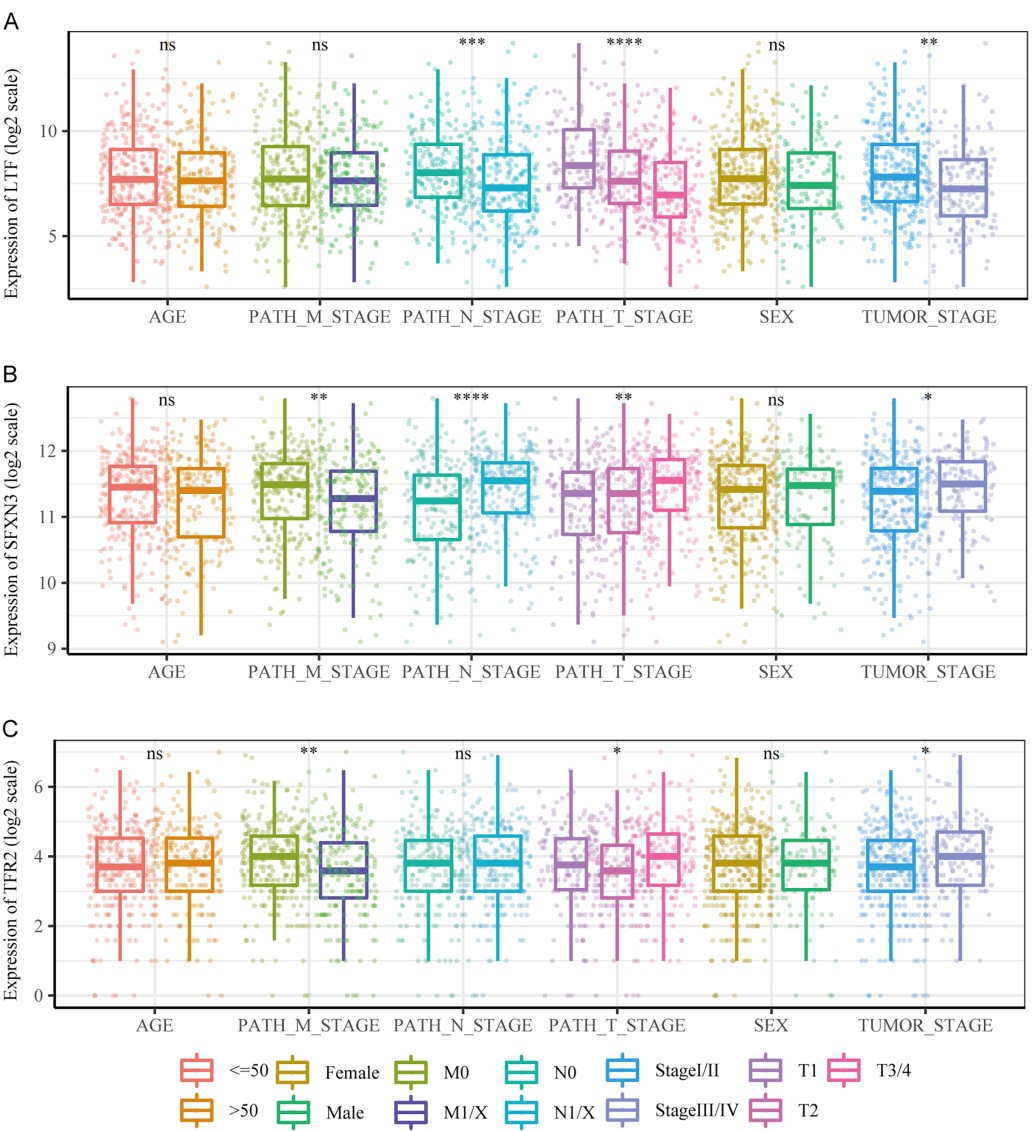

**Figure 8 Comparison of clinical characteristics of candidate prognostic markers.** (A–C) The boxplots of LTF, SFXN3, and TFR2 expression in different subgroups with different clinical characteristics, respectively. Wilcoxon or Kruskal-Wallis test was employed to evaluate variations between groups, and asterisks (*$P < 0.05$, **$P < 0.01$, ***$P < 0.001$, ****$P < 0.0001$) indicated statistically significant differences.

GEO validation set also verified this. Moreover, the TCGA validation set and the training set's survival analysis for risk grouping showed agreement (Figs. 7A–7C).

We compared the differences in candidate prognostic markers in various clinical subgroups. The outcomes indicated that there were LTF expression variations in subgroups of tumor stage and N and T stages, SFXN3 in subgroups of tumor stage and M, N, T stage, and TFR2 in subgroups of M, T stage and tumor stage (Figs. 8A–8C).

Subsequently, a multivariate prognostic model was established. Due to the correlation between the tumor stage and the TNM stage, we used a Cox regression model to build a multivariate prognostic prediction model according to the tumor, T, and N stages and RS.

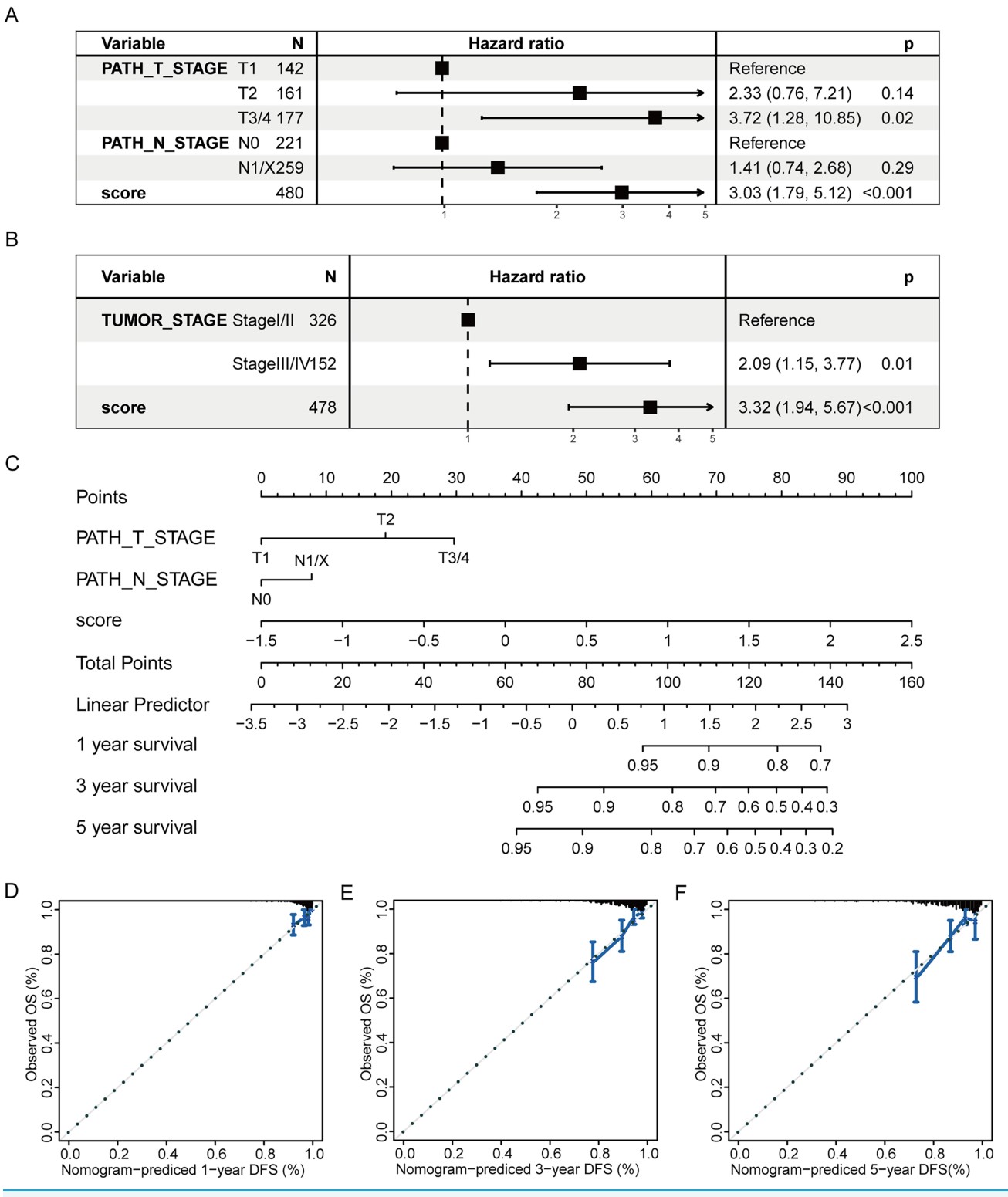

**Figure 9  Multivariate survival analysis of RS and construction of the prognostic model.** (A) The Cox model of T and N stages, in addition to RS, is presented in a forest map. (B) The Cox model of tumor stage and RS is presented in a forest map. (C–F) The nomogram and calibration curve of a multivariate Cox regression model for predicting patient survival with RS.

Figures 9A and 9B illustrate the forest map. The AIC value of the T and N stages, in addition to the RS model, was 487, while the AIC value of the tumor stage and RS model was 488. Therefore, the model of T and N stages, in addition to RS, was finally selected as the best model. Simultaneously, the nomogram (Fig. 9C) and calibration curve (Figs. 9D–9F) were drawn with an rms package to anticipate the 1-, 3- and 5-year survival possibility of individuals with PTC. The nomogram created with T and N stages (c-index = 0.67) and RS (c-index = 0.68) separately displayed that these variables could be good markers. However, the combined survival prediction model's c-index was 0.74, indicating a higher capacity for prognosis prediction.

## DEGs analysis and functional enrichment analysis in RS grouping

According to the high-/low- RS grouping, the Limma package of R language was employed to screen 465 DEGs from the TCGA mRNA expression matrix, including 289 mRNA down-regulated and 176 mRNA up-regulated. Figures 10A and 10B present the heatmaps of the DEGs and the volcano plot in the TCGA dataset.

KEGG analysis revealed that the mechanisms enriched by DEGs mainly included mineral absorption, tyrosine metabolism, and neuroactive ligand-receptor interaction (Figs. 10C–10F). The statistical outcome was shown in Table 3. GO analysis suggested that DEGs were mainly connected to BP, such as signal release, hormone transport, and regulation of membrane potential; CC, such as cation channel complex, synaptic membrane, and ion channel complex; and MF, such as ion channel, channel, and passive transmembrane transporter activities (Figs. 10G and 10H). The statistical outcome was shown in Table 4.

## Analysis of somatic mutation and immune cell infiltration

We subsequently analyzed the difference in somatic mutation among risk groups and the distribution of TMB. Compared to the low-risk group, the high-risk group had a significantly higher mutation proportion of BRAF (85% *vs* 56%) (Fig. 11A). In addition, compared to the low-risk group, the high-risk group also had a slightly higher TMB, and the significance level is at the critical value ($P = 0.054$) (Fig. 11B). According to the analysis of immune cell infiltration, the high-risk group exhibited a significantly higher degree of infiltration of immune cells, such as T cells regulatory, macrophages M0, and dendritic cells activated than the low-risk group ($P < 0.05$) (Fig. 11C).

The correlation analysis between the immune cell infiltration degree and the candidate genes' expression levels displayed that LTF was positively correlated with macrophages M1, T cells follicular helper, B cells memory, and B cells naïve, respectively ($P < 0.001$); SFXN3 was positively correlated with dendritic cells resting, macrophages M0, T cells regulatory, and T cells CD4 memory resting, respectively ($P < 0.001$), while it was negatively correlated with NK cells activated, T cells follicular helper, T cells CD8, and plasma cells, respectively ($P < 0.001$); TFR2 was positively correlated with T cells regulatory ($P < 0.001$), and negatively correlated with plasma cells ($P < 0.001$) (Fig. 11D).

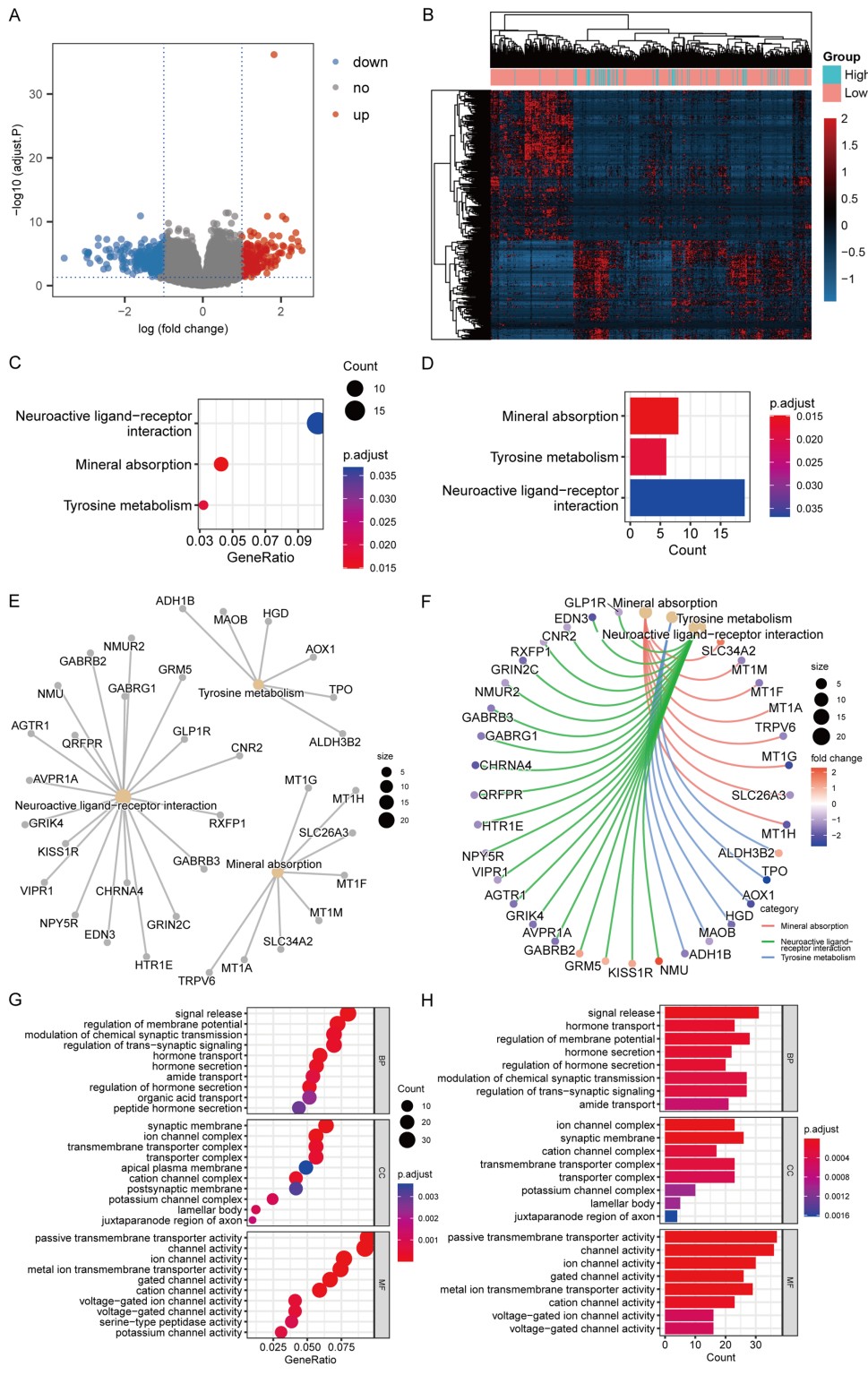

**Figure 10   DEGs analysis and functional enrichment analysis in RS grouping.** (A) Differential analysis volcano plot, blue dots for down-regulated, red dots for up-regulated. (B) Heatmap of differential analysis. (C) The bubble chart of KEGG analysis. The closer the color was to red, the smaller the *P* was, and the larger the bubble was, the more DEGs were enriched in this pathway. (D) The bar chart of KEGG analysis. The horizontal axis exhibited the number of genes enriched by the pathway, and the smaller the

**Figure 10** (continued)
P, the closer the color was to red. (E) The network diagram of KEGG analysis, displaying the relevant genes in the neuroactive ligand-receptor interaction, mineral absorption, and tyrosine metabolism pathways. (F) The network diagram of KEGG analysis, displaying the relevant genes in the neuroactive ligand-receptor interaction, mineral absorption, and tyrosine metabolism pathways and representing the fold change values of differential analysis in colors. Among them, red represented up-regulation, purple represented down-regulation, and darker colors indicated larger values. (G) The enrichment analysis of BP, CC, and MF by GO analysis is displayed in a bubble chart. More DEGs were enriched in this pathway. The closer the color was to red, the lower the P, and the bigger the bubble. (H) The enrichment analysis of BP, CC, and MF by GO analysis is displayed in a bar chart. The horizontal axis revealed the number of genes enriched by the pathway, and the lesser the P, the closer the color was to red.

**Table 3 Results of KEGG analysis of DEGs by RS grouping.**

| ID | Description | GeneRatio | BgRatio | P. adjust | Count |
|---|---|---|---|---|---|
| hsa04978 | Mineral absorption | 8/186 | 60/8,112 | 0.014858 | 8 |
| hsa00350 | Tyrosine metabolism | 6/186 | 36/8,112 | 0.018038 | 6 |
| hsa04080 | Neuroactive ligand-receptor interaction | 19/186 | 353/8,112 | 0.03683 | 19 |

**Table 4 Results of GO analysis of DEGs by RS grouping.**

| Ontology | ID | Description | GeneRatio | BgRatio | P. adjust | Count |
|---|---|---|---|---|---|---|
| BP | GO:0023061 | Signal release | 31/388 | 463/18,723 | 3.95E−05 | 31 |
| BP | GO:0009914 | Hormone transport | 23/388 | 306/18,723 | 0.00015 | 23 |
| BP | GO:0042391 | Regulation of membrane potential | 28/388 | 434/18,723 | 0.00015 | 28 |
| BP | GO:0046879 | Hormone secretion | 22/388 | 295/18,723 | 0.000199 | 22 |
| BP | GO:0046883 | Regulation of hormone secretion | 20/388 | 249/18,723 | 0.000199 | 20 |
| BP | GO:0050804 | Modulation of chemical synaptic transmission | 27/388 | 439/18,723 | 0.000292 | 27 |
| BP | GO:0099177 | Regulation of trans-synaptic signaling | 27/388 | 440/18,723 | 0.000292 | 27 |
| BP | GO:0042886 | Amide transport | 21/388 | 301/18,723 | 0.000636 | 21 |
| BP | GO:0015849 | Organic acid transport | 20/388 | 303/18,723 | 0.002329 | 20 |
| BP | GO:0030072 | Peptide hormone secretion | 17/388 | 236/18,723 | 0.002882 | 17 |
| CC | GO:0034702 | Ion channel complex | 23/408 | 282/19,550 | 1.09E−05 | 23 |
| CC | GO:0097060 | Synaptic membrane | 26/408 | 384/19,550 | 2.94E−05 | 26 |
| CC | GO:0034703 | Cation channel complex | 17/408 | 210/19,550 | 0.00026 | 17 |
| CC | GO:1902495 | Transmembrane transporter complex | 23/408 | 366/19,550 | 0.000273 | 23 |
| CC | GO:1990351 | Transporter complex | 23/408 | 381/19,550 | 0.000426 | 23 |
| CC | GO:0034705 | Potassium channel complex | 10/408 | 89/19,550 | 0.001003 | 10 |
| CC | GO:0042599 | Lamellar body | 5/408 | 17/19,550 | 0.001016 | 5 |
| CC | GO:0044224 | Juxtaparanode region of axon | 4/408 | 10/19,550 | 0.001626 | 4 |
| CC | GO:0045211 | Postsynaptic membrane | 17/408 | 277/19,550 | 0.003178 | 17 |
| CC | GO:0016324 | Apical plasma membrane | 20/408 | 367/19,550 | 0.00359 | 20 |
| MF | GO:0022803 | Passive transmembrane transporter activity | 37/390 | 481/18,368 | 8.59E−09 | 37 |
| MF | GO:0015267 | Channel activity | 36/390 | 480/18,368 | 1.69E−08 | 36 |

| | Table 4 (continued) | | | | | |
|---|---|---|---|---|---|---|
| Ontology | ID | Description | GeneRatio | BgRatio | P. adjust | Count |
| MF | GO:0005216 | Ion channel activity | 30/390 | 432/18,368 | 2.84E−06 | 30 |
| MF | GO:0022836 | Gated channel activity | 26/390 | 340/18,368 | 2.93E−06 | 26 |
| MF | GO:0046873 | Metal ion transmembrane transporter activity | 29/390 | 430/18,368 | 5.67E−06 | 29 |
| MF | GO:0005261 | Cation channel activity | 23/390 | 335/18,368 | 8.80E−05 | 23 |
| MF | GO:0005244 | Voltage-gated ion channel activity | 16/390 | 201/18,368 | 0.000491 | 16 |
| MF | GO:0022832 | Voltage-gated channel activity | 16/390 | 201/18,368 | 0.000491 | 16 |
| MF | GO:0005267 | Potassium channel activity | 12/390 | 121/18,368 | 0.000697 | 12 |
| MF | GO:0008236 | Serine-type peptidase activity | 15/390 | 191/18,368 | 0.000895 | 15 |

## Determination of IMRGs expression level and functional analysis in PTC

To determine the SFXN3 and TFR2 expression levels in PTC tissues, 40 PTC tissues and 38 non-tumorous thyroid tissues were detected, respectively. IHC (Fig. 12A) exhibited that the SFXN3 and TFR2 levels in PTC tissues were up-regulated. Next, to determine whether the elevated expression of SFXN3 and TFR2 was responsible for preserving the TC aggressive phenotype. SFXN3 and TFR2 were knocked down with high efficacy, confirmed at the mRNA levels, respectively (Fig. 12B). Proliferation was decreased in siSFXN3 or siTFR2 transfected TC cells, as demonstrated by a cell viability assay (Fig. 12C). These outcomes suggested that SFXN3 and TFR2 might function as oncogenes in PTC.

To explore the pathways of SFXN3 and TFR2 functioning in intracellular iron metabolism, a profile of iron-related protein mRNA expression was conducted, including ferritin (storage), ferroportin (FPN; iron exporter), divalent metal transporter 1(DMT1; iron importer) and TFR (iron importer). The results indicated that the silence of SFXN3 or TFR2 significantly increased TFR expression while decreasing FTL and FTH expression (Fig. 12D). Further WB experiments also confirmed this (Fig. 12E).

## DISCUSSION

Although most PTC patients have ideal postoperative effects and good prognoses, some well-differentiated TC tumors are still more invasive (Wen et al., 2021). They are resistant to standard treatments, such as those that are non-operable, relapse after surgery, and do not respond to radioiodine treatment (Qin et al., 2021). Additionally, PTC is a very heterogeneous condition, and the development of tumors includes a complicated network made up of several signaling mechanisms (Balachandran et al., 2015; Zeiger & Schneider, 2013). Therefore, it is particularly important to detect markers related to tumor diagnosis and prognosis to formulate treatment and follow-up plans.

Iron is vital for cell viability as it exists in proteins that perform diverse functions, including respiration, transport, and homeostasis of oxygen, as well as the synthesis of biomolecules (Dlouhy & Outten, 2013). Moreover, iron acts as a crucial component of many proteins, including the repair and metabolism of nucleic acid as well as the progression of the cell cycle (Zhang, 2014). Since iron is an essential part of physiology and

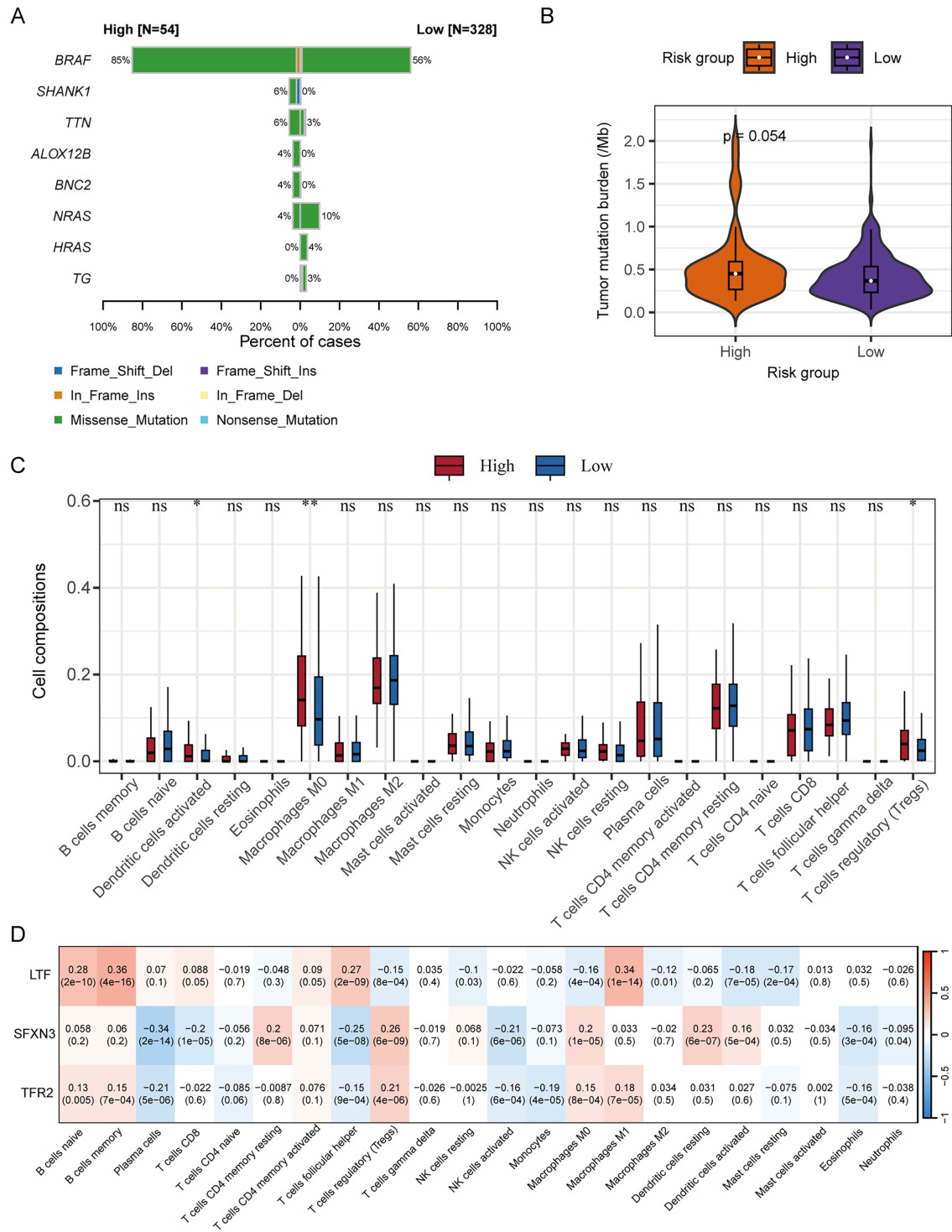

**Figure 11 Correlation analysis of RS grouping and somatic mutation and immune cell infiltration of candidate genes.** (A) Genes with different somatic mutation proportions between high- and low-risk scores groups. (B) Comparison of TMB between high- and low-risk scores groups. Differences between groups were assessed by the Wilcoxon test. (C) Comparison of immune cell infiltration degree between high- and low-risk scores groups. Differences between groups were assessed by the Wilcoxon test, and asterisks (*P < 0.05, **P < 0.01) indicated statistically significant differences. (D) Correlation matrix between candidate gene expression and degree of immune cell infiltration. Positive correlations were represented by red, negative correlations by blue, and darker colors indicated a stronger correlation. A grid within the matrix displayed the correlation coefficient and P-value.

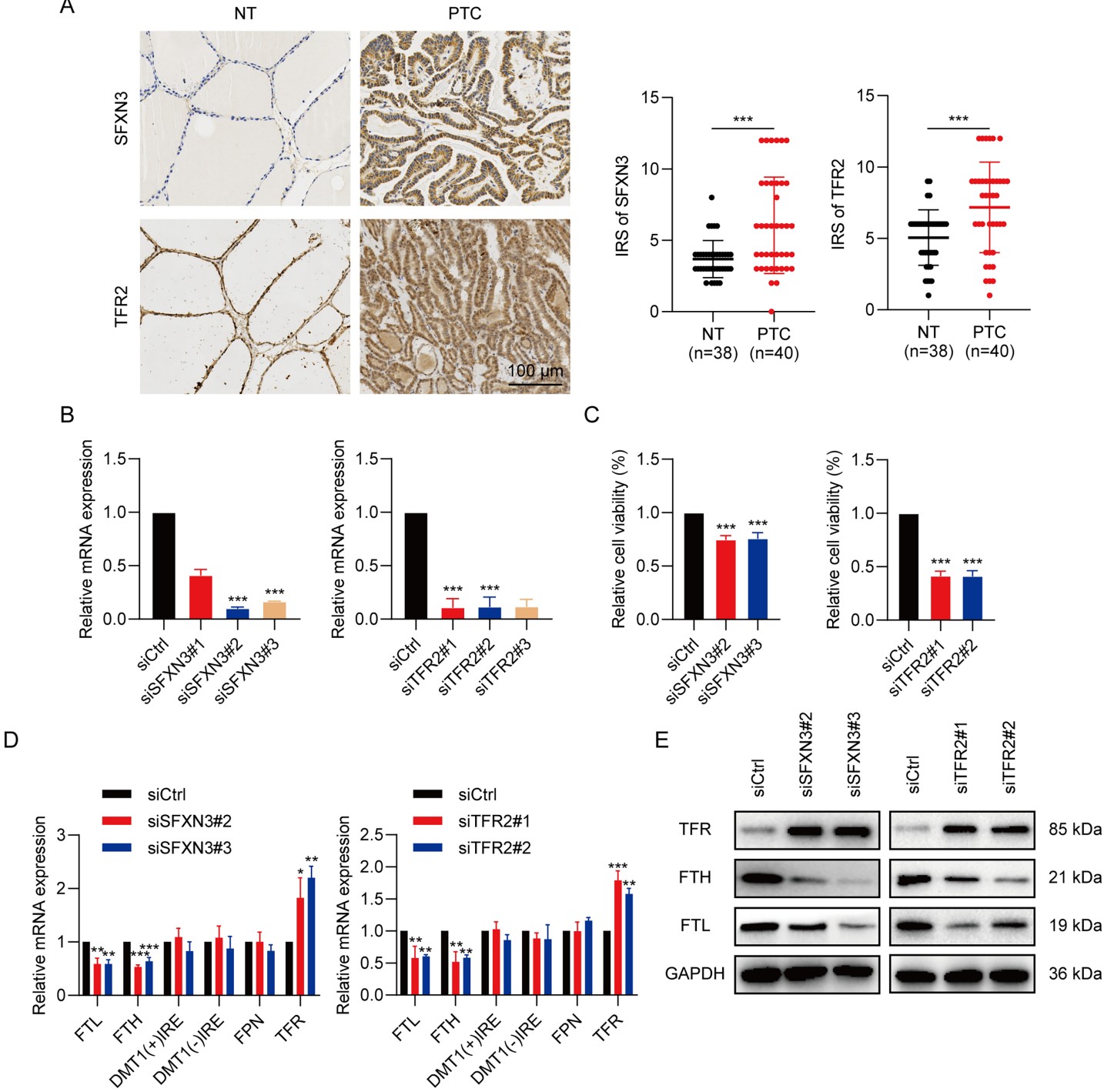

**Figure 12 IMRGs expression verification and functional analysis.** (A) In normal and PTC tissues, IHC staining was employed to analyze SFXN3 and TFR2 expression. (B) Verification of SFXN3-siRNA and TFR2-siRNA silencing efficacy at the mRNA level in TPC-1 cells. (C) Cell viability assay was measured after SFXN3 or TFR2 silencing. (D-E) mRNA and protein levels of iron-related proteins expression after SFXN3 or TFR2 silencing. DMT1 (−) IRE, divalent metal transporter 1 without iron responsive element; DMT1 (+) IRE, divalent metal transporter 1 with iron responsive element. $^{*}P < 0.05$, $^{**}P < 0.01$, $^{***}P < 0.001$.

anatomy with low bioavailability, the human body regulates iron stores strictly to maintain conservation and minimize toxic effects (*Dev & Babitt, 2017*). The iron metabolism imbalance contributes to many diseases, and it is significant to explore this role in the search for therapeutics (*Kell, 2009*).

Many recent studies have demonstrated that iron metabolism has a function in all cancer development stages (*Guo et al., 2021*; *Recalcati, Gammella & Cairo, 2019*). The total disturbance of iron absorption, effusion, storage, and modulation in malignancies suggests that reprogramming iron metabolism will result in the dysregulation of proliferation and survival in tumor cells (*Andrews, 2008*; *Jung et al., 2019*; *Manz et al., 2016*; *Wang et al., 2018*). Recent investigations have shown that vitamin C activates ferritinophagy to cause ferroptosis in anaplastic thyroid tumor cells (*Wang et al., 2021*). *Lin et al. (2021)* also used network analysis to examine the prognosis of PTC and ferroptosis genes in immune infiltration. The dynamic profile of genes associated with iron metabolism in TC is still unknown.

In this investigation, to screen seven down-regulated and five up-regulated IMR-DEGs, we employed clinicopathological data and gene expression from the open-access database. Network analyses indicated that these genes were associated with multiple transcription factors, miRNAs, small molecular compounds, and drugs, which may reveal their role in tumor progression and potential as therapeutic targets. Subsequently, we used Cox and LASSO regression analyses to create three predictive IMRGs signatures and established an RS system to construct and evaluate a nomogram for predicting prognosis.

Using LASSO regression and univariate Cox analyses, a total of three IMR-DEGs were found to be possible predictive biomarkers, and these genes were utilized to build a predictive model. Two of them (SFXN3, TFR2) had negative correlations between their expression levels and DFS, but the expression level of the LTF gene correlated positively with DFS. During one-carbon metabolism, SFXN3 is a crucial mitochondrial serine transporter that is involved in tumor cell growth (*Chen et al., 2022*). Recent research has shown that the immunosuppressive microenvironment was linked to the elevated SFXN3 expression mediated by non-coding RNA, which was a predictive marker in head and neck squamous cell cancer (HNSC) (*Chen et al., 2022*). Through performing a competitive endogenous RNA analysis, *Zheng et al. (2022)* also showed that MIR193HG-miR-29c-3p-SFXN3 participated in HNSC, which significantly influenced the treatment efficacy and prognosis. In addition, *Murase et al. (2008)* indicated that the serum SFXN3 autoantibodies may act as a new cancer biomarker for oral squamous cell cancer. Consistent with these studies, our research also indicated that the elevated SFXN3 expression represented a poor predictive marker and an oncogene in PTC, and its silence significantly inhibited the proliferation of PTC cells.

Iron is transported to cells from peripheral blood through a transmembrane glycoprotein receptor family called the transferrin receptor family. TFR2, a subtype of TFRs, has been found altered expression in tumor cells. An elevated TFR2 expression was identified in the colon, glioblastoma (GBM), and ovarian cancer cell lines (*Calzolari et al., 2009*; *Calzolari et al., 2010*). However, in subjects with myelodysplastic syndrome, acute myeloid leukemia, and GBM, the higher transcription levels of TFR2 were connected to a

better prognosis than that in the lower group (*Calzolari et al., 2010*; *Di Savino et al., 2017*; *Nakamaki et al., 2004*). The phenomenon might be due to the fact that TFR2 sensitizes tumor cells to cell-cycle-specific chemotherapy medications by regulating potential cellular signaling (*Zhao et al., 2020*). For instance, the increased TFR2 expression in GBM cells led to a high increase in growth, which was highly sensitive to temozolomide (*Calzolari et al., 2010*). In our work, we illustrated that the increased TFR2 expression is related to a worse prognosis and proliferation of PTC, and this phenomenon is different from that of other tumors. Exactly, it will be of potential value to explore the internal mechanism behind these differences in the future.

LTF is an iron-binding transport glycoprotein identified as a key immune-related gene correlated with PTC prognosis (*Liu et al., 2021*; *Qin et al., 2021*). LTF expression level in cancer tissue was markedly lower than that in healthy tissue. Also, ssGSEA outcomes indicated that LTF expression level was strictly connected to the immune process (*Qin et al., 2021*). This is consistent with current results that LTF was downregulated in PTC and was related to a better prognosis. Furthermore, *Hosseinkhan et al. (2020)* determined LTF as a promising biomarker down-regulated in a stage I subgroup and in the common stage IV of PTC. Taken together, this evidence suggests that these genes may have a vital function in the incidence and progression of malignancies and might be ideal prognostic markers of PTC prognosis.

The multigene signature for DFS was an independent predictive variable in PTC patients in subsequent studies. Patients with high risk possessed a worse DFS than those with low risk, according to risk classification by RS. This model was successfully and consistently verified in several patient populations, and multivariate Cox regression analysis validated its independence as a predictive marker. To further predict the DFS of individuals with PTC, we built a predictive nomogram model depending on IMRGs, which included the T and N stages in addition to RS. The calibration curves also demonstrated the nomogram's reliable prognostic value for DFS in the TCGA cohort. This nomogram model could be utilized to examine PTC patients' prognoses and schedule follow-up plans. Interestingly, we observed that compared to the low-risk group, the high-risk group had significantly higher BRAF mutation frequency, TMB, and some immune cell infiltration. BRAF mutation is a common event in PTC, and its presence indicates tumor progression (*Guerra et al., 2014*; *Kim et al., 2014*). Moreover, the amount of tumor somatic coding mutations is defined as TMB, which is used to estimate neoantigen burden and to predict immune checkpoint inhibition response across a variety of tumor types (*Sha et al., 2020*). Overall, these findings indicated the clinical significance of RS grouping beyond predicting prognosis.

The clinical samples were subsequently validated by IHC, which revealed elevated SFXN3 and TFR2 levels in cancerous tissue. It is well known that iron is essential for ribonucleotide reductase's catalytic activity, an enzyme responsible for converting ribonucleotides into deoxyribonucleotides, the rate-limiting stage of DNA synthesis as well as a necessary step in cell replication (*Torti & Torti, 2020*). Massive hepcidin release is a TC biomarker that causes reduced iron exporter, FPN, expression, and elevated intracellular iron retention, thus promoting tumor growth (*Zhou et al., 2018*). Therefore, we conducted

the cell viability assay to determine the silencing IMRGs' effect on the cell proliferation ability of PTC cell line TPC-1. The outcomes exhibited that both SFXN3 and TFR2 knockdown suppressed the TC growth *in vitro*. Notably, in our study, SFXN3 and TFR2 could affect iron metabolism through the regulation of FTL, FTH, and TFR expression in iron-related proteins. FTL and FTH belong to ferritin, while TFR belongs to the iron importer. After silencing SFXN3 or TFR2, FTL, and FTH were down-regulated, while TFR was up-regulated, all of which could up-regulate intracellular iron concentration. These results suggested that the silence of SFXN3 or TFR2 may lead to the overload of intracellular $Fe^{2+}$, thereby causing ferroptosis (*Liu et al., 2022*; *Lu et al., 2021*). Collectively, these investigations also suggest that SFXN3 and TFR2 may be viable therapeutic targets and markers of clinical prognosis in PTC.

However, there are still restrictions on our work. First, this was a retrospective work with bias in selecting variables such as clinicopathological features. Therefore, there is still a need to improve the accuracy of the data. Second, our prediction model depends on the survival function estimation after analyzing multiple influencing factors in TCGA comprehensively. Hence, there are limited clinical implications from this study. It is essential to obtain more prospective information to validate its clinical significance. Finally, although we revealed the characteristics of three IMRGs, it is still crucial to carry out further trials to clarify the roles and pathways in PTC development.

## CONCLUSIONS

In conclusion, this study provided, as far as we know, the first application of IMRGs to evaluate their prognostic prediction ability in PTC. We examined IMRGs in PTC and found three genes linked to the predictive and clinicopathological features of PTC, of which two were validated in 40 PTC and 38 NT samples by IHC, and conducted *in vitro* experiments to explore the biological function. Additionally, we created and verified an RS system for risk categorization and prognosis. Finally, a nomogram model was built that demonstrated high predictive accuracy for 1-, 3- and 5-year DFS rate predictions. These findings indicate that IMRGs possess the predictive ability of PTC prognosis, and our prognostic model could be included in large-scale prospective research in the future to further verify its clinical value.

### Funding
Our research was funded by the National Natural Science Foundation of China under Grant (U20A20382). The funders had no role in study design, data collection and analysis, decision to publish, or preparation of the manuscript.

### Grant Disclosures
The following grant information was disclosed by the authors:
National Natural Science Foundation of China: U20A20382.

## Competing Interests

The authors declare that they have no competing interests.

## Author Contributions

- Tiefeng Jin conceived and designed the experiments, performed the experiments, authored or reviewed drafts of the article, and approved the final draft.
- Luqi Ge analyzed the data, prepared figures and/or tables, authored or reviewed drafts of the article, and approved the final draft.
- Jianqiang Chen performed the experiments, prepared figures and/or tables, and approved the final draft.
- Wei Wang conceived and designed the experiments, analyzed the data, prepared figures and/or tables, authored or reviewed drafts of the article, and approved the final draft.
- Lizhuo Zhang conceived and designed the experiments, prepared figures and/or tables, authored or reviewed drafts of the article, and approved the final draft.
- Minghua Ge conceived and designed the experiments, authored or reviewed drafts of the article, and approved the final draft.

## Human Ethics

The following information was supplied relating to ethical approvals (*i.e.*, approving body and any reference numbers):

Ethics Committee approval was obtained from Zhejiang Provincial People's Hospital for all studies (Ethical Approval No: 2021QT251).

## Data Availability

The raw data is available at figshare:

Jin, Tiefeng; Ge, Luqi; Chen, Jianqiang; Wang, Wei; Zhang, Lizhuo; Ge, Minghua (2023): RAW_DATA. figshare. Journal contribution. https://doi.org/10.6084/m9.figshare.22267813.v1.

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
