# Peer review of "Identification of iron metabolism-related genes as prognostic indicators for papillary thyroid carcinoma: a retrospective study"

_PeerJ, doi:10.7717/peerj.15592_

## Round 0.1 · original submission · Major Revisions

Please revise the manuscript according to the reviewers' suggestions.

Reviewer 1 ·

Basic reporting

The manuscript is generally easy to understand. The article structure is fine. Raw data shared.
However, the clinical significance of this study is limited.

Experimental design

1. The differences of cellular mutation distribution and tumor mutation burden (TMB) across different iron metabolism models are suggested to be analyzed. In addition, the correlation between IMRGs and other immune invasion characteristics is suggested to be analyzed.

Validity of the findings

1. In order to further discover the prognostic value of IMRG, the expression of SFXN3 or TFR2 in collected clinical samples and their correlation with clinical features such as TNM stage, histological type, and immune invasion characteristics are suggested to be analyzed.
2. Cell viability assay was measured after SFXN3 or TFR2 silencing in this study. In addition to cell viability detection, the expression of key proteins in related pathways is suggested to be analyzed after knocking down SFXN3 and TFR2, as well as the response changes to drugs after knockout.

Reviewer 2 ·

Basic reporting

In this study Jin et al. investigated the prognostic role of the iron metabolism related genes in the papillary thyroid carcinoma. Overall the study is well reported and results were documented.

Major concerns:
The results of Pearson correlation analysis and its related figure (figure 3) was not explained in the text.

Experimental design

Authors have used the existing thyroid cancer datasets and applied several computational and statistical methods to determine whether IMRG expression pattern is a risk factor or clinical prognosis in papillary thyroid cancer patients. The authors have clearly stated experimental design.

Validity of the findings

Overall the results and methods are well stated and support the finding that the IMRGs can be used as prognostic marker in the PTC

·

Basic reporting

The manuscript by Jin and coworkers titled “Identification of iron metabolism-related genes as prognostic indicators for papillary thyroid carcinoma” describes 3-gene signature that predicts prognosis of papillary thyroid carcinoma. This is an excellent study, and the sample size is adequate. The prognosis was trained and validated in independent sets of samples. The study design is appropriate for the research question. The introduction is written well, and the methods were clearly described. Results were clearly explained in the text and adequately supported by the figures and tables. The discussion is well written. However, there are a few minor issues with this manuscript, and I ask the authors to address these issues in the revision.
1. In figure 5A, there were 3 genes in red color font (probably there are statistically significant in univariate analysis). Please show the p values in this figure. This is important because these genes were further used as a panel due to their statistical significance.
2. There is no need for LASSO analysis shown in figure 5C and D. LASSO model was constructed using 3 genes and it could not further reduce the feature size. All the genes were needed to have an effective model. Therefore, LASSO failed to create a sparse model which was intended to create. I suggest replacing this model with a Cox regression model which is clinically relevant. In addition, such a model would be identical to the calibration curves shown in figure 9D.
3. I ask authors to elaborate figure legends. The legends should have all the details to understand and interpret all the components shown in the figure. I am not listing each such instance across all the figures in this report. For example, figure 11 E and F have KEGG pathway identifiers as numbers. Reader would not understand what these pathways are. Either replace these numbers with pathway names or list the names in the legend.
4. Figure 4C, D, E, F and G: The font size of these figures is not legible. Please increase the font size.

Experimental design

No comment

Validity of the findings

No comment

---

## Round 0.2 · accepted · Accept

Since the authors have addressed the reviewers' concerns, this manuscript can be accepted now.

Reviewer 1 ·

Basic reporting

The authors have well addressed the comments.

Experimental design

The authors have well addressed the comments.

Validity of the findings

The authors have well addressed the comments.

·

Basic reporting

The authors addressed my comments.

Experimental design

-

Validity of the findings

-